# Selective electrochemical oxidative coupling of methane mediated by $Sr_2Fe_{1.5}Mo_{0.5}O_{6-\delta}$ and its chemical stability

Kannan P. Ramaiyan [1], Luke H. Denoyer[1], Angelica Benavidez[1] & Fernando H. Garzon [1]✉

Efficient conversion of methane to value-added products such as olefins and aromatics has been in pursuit for the past few decades. The demand has increased further due to the recent discoveries of shale gas reserves. Oxidative and non-oxidative coupling of methane (OCM and NOCM) have been actively researched, although catalysts with commercially viable conversion rates are not yet available. Recently, $Sr_2Fe_{1.5+0.075}Mo_{0.5}O_{6-\delta}$ (SFMO-075Fe) has been reported to activate methane in an electrochemical OCM (EC-OCM) set up with a C2 selectivity of 82.2%[1]. However, alkaline earth metal-based materials are known to suffer chemical instability in carbon-rich environments. Hence, here we evaluated the chemical stability of SFMO in carbon-rich conditions with varying oxygen concentrations at temperatures relevant for EC-OCM. SFMO-075Fe showed good methane activation properties especially at low overpotentials but suffered poor chemical stability as observed via thermogravimetric, powder XRD, and XPS measurements where $SrCO_3$ was observed to be a major decomposition product along with $SrMoO_3$ and MoC. Nevertheless, our study demonstrates that electrochemical methods could be used to selectively activate methane towards partial oxidation products such as ethylene at low overpotentials while higher applied biases result in the complete oxidation of methane to carbon dioxide and water.

[1] Center for Micro-Engineered Materials, Department of Chemical and Biological Engineering, University of New Mexico, Albuquerque, NM 87106, USA.
✉email: Garzon@unm.edu

Converting methane into value-added products such as ethylene, propylene, benzene, methanol, etc. in an economical and environmentally friendly manner remains a grand challenge in chemistry. Methane is quite stable, has strong C-H bonds (first bond dissociation energy—439.3 kJ mol$^{-1}$), and is difficult to activate under normal conditions[2]. Methane conversion has become especially important due to recent discoveries of large shale gas reserves, drastically lowering methane's price[2–4]. Among the olefins, methane conversion to ethylene ($C_2H_4$) is considered potentially more productive than other products due to ethylene's utility in a wide variety of industries, such as chemical feedstock and building block. Ethylene's worldwide consumption exceeded 150 million tons in 2017, further exemplifying its value[5]. Ethylene is currently produced through the steam cracking of methane and naphtha at high temperatures (>750 °C), which is a high-energy and carbon-intensive process that releases 1–2 tons of $CO_2$ per ton of ethylene produced[5]. Hence, a direct conversion of methane to ethylene without the multi-step process could be more economical. This could be achieved by an oxidative coupling or non-oxidative coupling of methane (OCM and NOCM, respectively), although the NOCM process suffers from highly unfavorable thermochemistry[6,7]. OCM pioneered by Keller and Bhasin, on the other hand, suffers from overoxidation to $CO_2$ and $H_2O$ as the partial oxidation products of methane such as ethane and ethylene are more active than methane thus creating selectivity challenges[8].

Electrochemical oxidative coupling of methane (EC-OCM) to ethylene in solid oxide electrolyzers (SOEs) is attracting attention recently since the methane activation could be regulated not only by just oxide ions but also by the application of potentials coupled with the ability to utilize renewable electrical energy in remote areas[1,9–11]. Oxide ion-conducting SOEs, where oxygen is transported from cathode to anode as oxide ions, normally operate at temperatures >800 °C, which are suitable for the catalytic conversion of methane to ethylene. Recently, $Sr_2Fe_{1.5+x}Mo_{0.5}O_{6-\delta}$ (strontium Iron molybdate (SFMO)) was successfully demonstrated for electrochemical conversion of methane to ethylene where the role of Fe nanoparticles exsolved in situ under reducing atmospheres in SFMO was evaluated. A small excess of Fe from the stoichiometric levels produced Fe nanoparticles (~25 nm) imbedded on the metal oxide scaffold. These nanoparticles significantly enhanced the catalytic activity while also resisting coke formation and showed a durable performance for 100 h[1]. An overall $C_2$ selectivity of 81.2% was achieved for $Sr_2Fe_{1.5+0.075}Mo_{0.5}O_{6-\delta}$ (SFMO-075Fe) as anode at an applied potential of 1.6 V. However, one of the major challenges with OCM, under considerable oxygen flux, is the formation of overoxidation products such as $CO_2$ and water, which was not observed with the generic five tested SFMO catalysts. Similarly, materials containing alkaline earth metals such as Ba and Sr are prone to form their carbonates in carbon-rich environments at elevated temperatures, which also was not observed in this work[12,13]. Our HSC Chemistry (HSC Chemistry version 7.193 from Outotec®) calculations for the reaction between SrO and $CO_2$ indicate $SrCO_3$ as the preferred product till 1200 °C ((for HSC Chemistry data, see Supplementary Fig. 1). Hence, in this report, we evaluate the SFMO-075Fe catalyst for EC-OCM in a button cell set-up identical to the reported set-up operating at 850 °C. The catalyst was characterized by impedance measurements, cyclic voltammetry (CV), and chronoamperometric (CA) measurements in a wide potential window to better understand the role of applied bias. Evolved gas analysis was performed using mass spectroscopy and gas chromatography. Electrochemical measurements were carried out utilizing cathode as the reference electrode with an assumption that the electrode potential is close to the thermodynamic potential for an oxygen electrode. CA measurements indicate that ethylene is selectively produced at low overpotentials while high overpotentials lead to the formation of $CO_2$ and $H_2O$. CV measurements demonstrate a preferred potential window for ethylene production. Chemical stability measurements on SFMO indicate the formation of carbonates in pure methane and in 10% oxygen in methane mixtures as anticipated. Our results indicate the possibility of EC-OCM to produce ethylene selectively at low overpotentials along with cogeneration of energy.

## Results and discussion

**EC-OCM measurements.** Electrochemical impedance Nyquist plots obtained under various bias potentials at 850 °C in pure methane are given in Supplementary Fig. 2 while mass spectroscopic data observed on the outlet gas stream is given in Supplementary Fig. 3. Clearly, application of an electrochemical potential induces an increased ethylene concentration in the product stream along with an increase in hydrogen, concurrently. This clearly illustrates the SFMO-075Fe's ability to activate methane to produce ethylene. However, a further increase in applied potentials and subsequent higher currents results in the observation of $H_2O$ and $CO_2$ along with a corresponding reduction in ethylene and hydrogen concentrations. This behavior clearly indicates the overoxidation of methane under a higher flux of oxide ions across the electrolyte at higher applied potentials. The high frequency intercept on the $X$ axis resistance remained close to 2.75 ohm for applied potentials from −1.00 V to 0.00 V. However, further increase in applied bias resulted in increase of this value and reached 3.75 ohm for 1.00 V bias. Nevertheless, the charge transfer resistance values continually decreased with increasing bias values indicating a faster reaction kinetics that may support the complete oxidation of methane to $CO_2$ and $H_2O$. To understand the role of applied bias further, we carried out CV measurements at a slow scan rate of 1 mV s$^{-1}$ in a wide potential range (−1.8 to 1.6 V). The results presented in Fig. 1 show a comparison of potential, current, and concentration variation of the four products ($C_2H_4$, $H_2$, $CO_2$, and $H_2O$) observed on the outlet stream as a function of time. It is evident from Fig. 1 that there are two potentials at which the ethylene production reaches a peak value: about −0.75 and −0.25 V. The peak at −0.75 V is observed during the cathodic cycle under significantly lower current densities ~50 mA cm$^{-2}$. The ethylene peak at −0.25 V is observed at a current density of ~200 mA cm$^{-2}$ and coincides with the initiation of $H_2O$ and $CO_2$ peaks in the mass spectroscopy. Ethane formation was not detected. The evolution of $H_2O$ and $CO_2$ demonstrates that higher biases and current densities (potentials >0.00 V) result in overoxidation products instead of the desired partial oxidation product, ethylene. Both $H_2O$ and $CO_2$ concentrations hit the maximum at the maximum applied potential of 1.60 V, which is also associated with maximum observed current densities. Electrochemical oxidation of $CH_4$ using $O^{2-}$ ions is considered to be a charge rebalancing process that involves the transformation of $O^{2-}$ ions on the active interface to different types of oxygen species, including $O^-$ and $O^{2-}$ [14]. Among the oxygen species, $O^-$ is reported as the best candidate for activating methane especially toward partial oxidation products such as ethylene. At high current densities, the charge rebalancing may not be effective due to higher flux of oxide ions, which could result in $O^{2-}$ ions either interacting directly with $CH_4$ or forming $O_2$ molecule that reacts further with $CH_4$ both of which would produce predominantly $CO_2$ and $H_2O$[14,15]. The nature of oxygen source is also known to play a role on the product selectivity as $N_2O$ was reported to produce ethylene more selectively than pure oxygen[16]. Similarly, OCM process under low oxygen concentration has been reported to produce ethylene and other partial oxidation products such as ethane and ethylene. However, at high oxygen concentrations in the reaction mixture, the

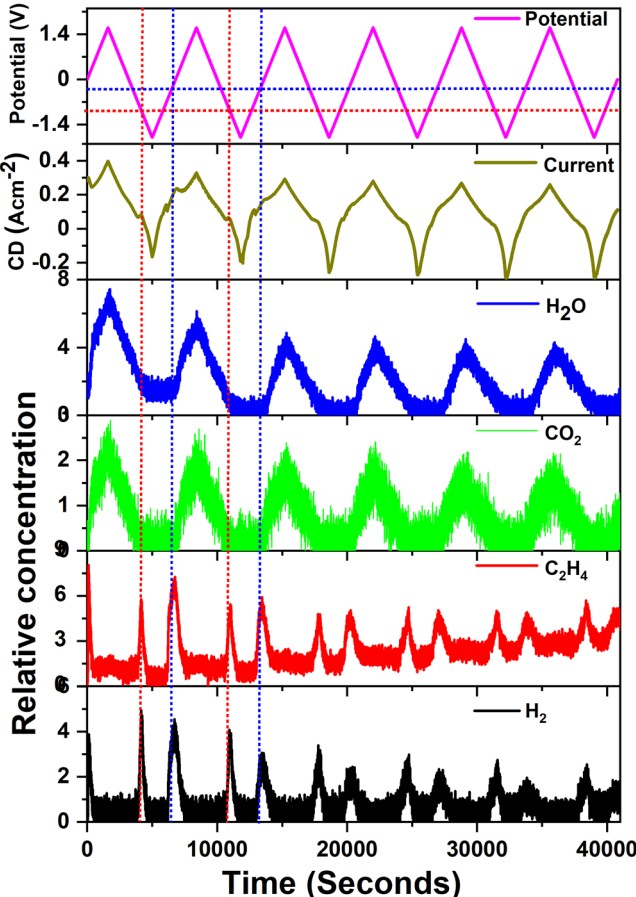

**Fig. 1 Cyclic voltammetric measurements coupled with mass spectroscopic data.** Potential and current applied during cyclic voltammetric measurements as a function of time is presented along with the variation in concentration for the four methane oxidation products obtained on the outlet stream demonstrating the correlation between applied bias and methane activation.

product stream tends to have more $CO_2$ and $H_2O$[17]. Both of this could explain the rise in $CO_2$ and $H_2O$ in our EC-OCM measurements as at higher applied biases the oxide ion flux is increased, associated with high currents (~300 mA cm$^{-2}$—Fig. 1)[14,15]. Figure 1 also presents a reduction in current density and in the concentration of the methane conversion products with increasing cycle number that implies a direct correlation between the current and methane conversion products. However, it also indicates the instability of the catalyst to sustain its catalytic activity. The individual CV cycles (for CV measurements, see Supplementary Fig. 4) show a drop in peak intensity for the peak near −0.75 V in concurrence with a reduction of ethylene production and hence could be associated directly with the electrochemical methane activation. Nevertheless, these results clearly display a preferred potential window, between −0.75 and −0.25 V, for selective conversion of methane to ethylene in EC-OCM experiments. Further, the CV measurement coupled with mass spectroscopic analysis also indicated no ethylene production at potentials <−1.0 V ruling out the possibility of NOCM processes for this catalyst (for common reactions involved in conversion of methane to ethylene, see Supplementary Note 1 and Supplementary Eqs. (1–8)).

**CA measurements**. CA measurements on the button cell were carried out by maintaining the cell initially at −1.25 V for 300 s to

ensure zero current and no EC-OCM as inferred from CV measurements. Following this, various bias potentials (from −1.00 to 1.20 V), where a positive current was observed in CV measurements, were applied for 900 s and the results are shown in Supplementary Fig. 5. The current density was increasing with the application of applied biases from −1.0 to −0.5 V and reached steady-state values quickly. However, at higher biases, the initial high currents were not sustained, and within 200 s, the current density dropped to a lower steady-state current. The mass spectra analyses obtained on the gas outlet stream during these CA measurements are given in Fig. 2. A non-zero positive current has resulted in the observation of ethylene and hydrogen. The concentration associated with these two products increased linearly up to −0.50 V. However, further increase in applied bias affected the product selectivity as $H_2O$ and $CO_2$ begin to appear in the product stream at the expense of ethylene and hydrogen. At applied potentials above 0.75 V the initial rise in ethylene concentration has fallen back close to background levels within 300 s of operation while $CO_2$ and $H_2O$ reached their maximum values. This is in agreement with the CV and impedance measurements where at higher applied biases, the higher $O^{2-}$ ions flux due to higher current densities result in complete oxidation[15]. Interestingly, after 900 s (point 3 on Fig. 2i), when the bias potential was removed, there is another sharp rise in mass spectra data for mass 28 (ethylene + CO) and hydrogen concentration that could be due to the residual flux of incoming oxide ions. Possible reactions involved in EC-OCM process are given in Supplementary Eqs. (3–8) (for OCM reactions, see Supplementary Note 1) where the production of 1 mole of ethylene leads to 1 mole of $H_2$ (Supplementary Eq. (3)), whereas 1 mole of CO leads to 2 moles of $H_2$ (Supplementary Eq. (5)). The mass spectrum peak for mass 28 at point 3 is associated with roughly twice the rise in hydrogen intensity providing more evidence that CO could be the dominant product at this point under higher oxygen flux with no potential bias applied. This further underlines the influence of the higher applied biases that may have played a role on the over-oxidation of methane to CO and $CO_2$. $H_2O$ is not observed at low applied biases in contrast to Supplementary Eqs. (3) and (4) possibly due to condensation in the lines. Interestingly, we did not observe any significant quantities of ethane in the outlet stream on the mass spectroscopic analysis. HSC thermochemical modeling of the reaction Gibbs free energies between methane and oxygen molecule indicate that ethylene is the preferred product at 850 °C by 80 kJ mol$^{-1}$ (for HSC calculations on methane conversion, see Supplementary Fig. 6a). This observation agrees well with EC-OCM reports at this temperature where ethylene is observed as the major product[10,11]. Interestingly, the cell electromotive force values calculated from the HSC calculations (Supplementary Fig. 6b) indicate that, at an operating temperature of 850 °C, the equilibrium cell potential for producing ethylene is in the range of −0.8 to −0.6 V against oxygen reference electrode. This agrees well with our observation as the maximum ethylene production was observed in this range.

Since the ethylene production varies significantly in the CA measurements from the initial peak value at about 400 s to the bias removal point at 1200 s (specifically at high applied biases), we calculated the faradaic efficiency (FE) of the EC-OCM process at these two extreme points (point 1 and 2 as indicated in Fig. 2i) for all applied biases. Observed FE plots with respect to ethylene and $CO_2$ at these specific points are given in Supplementary Fig. 8 (for FE plots, see Supplementary Fig. S8). Ethylene production by EC-OCM is generally considered as a four-electron process as given in Supplementary Eq. (4). However, since we observed hydrogen in the product stream, we calculated FE as per Supplementary Eq. (3), a two-electron process. According to this, a maximum of 40% FE was obtained at very low

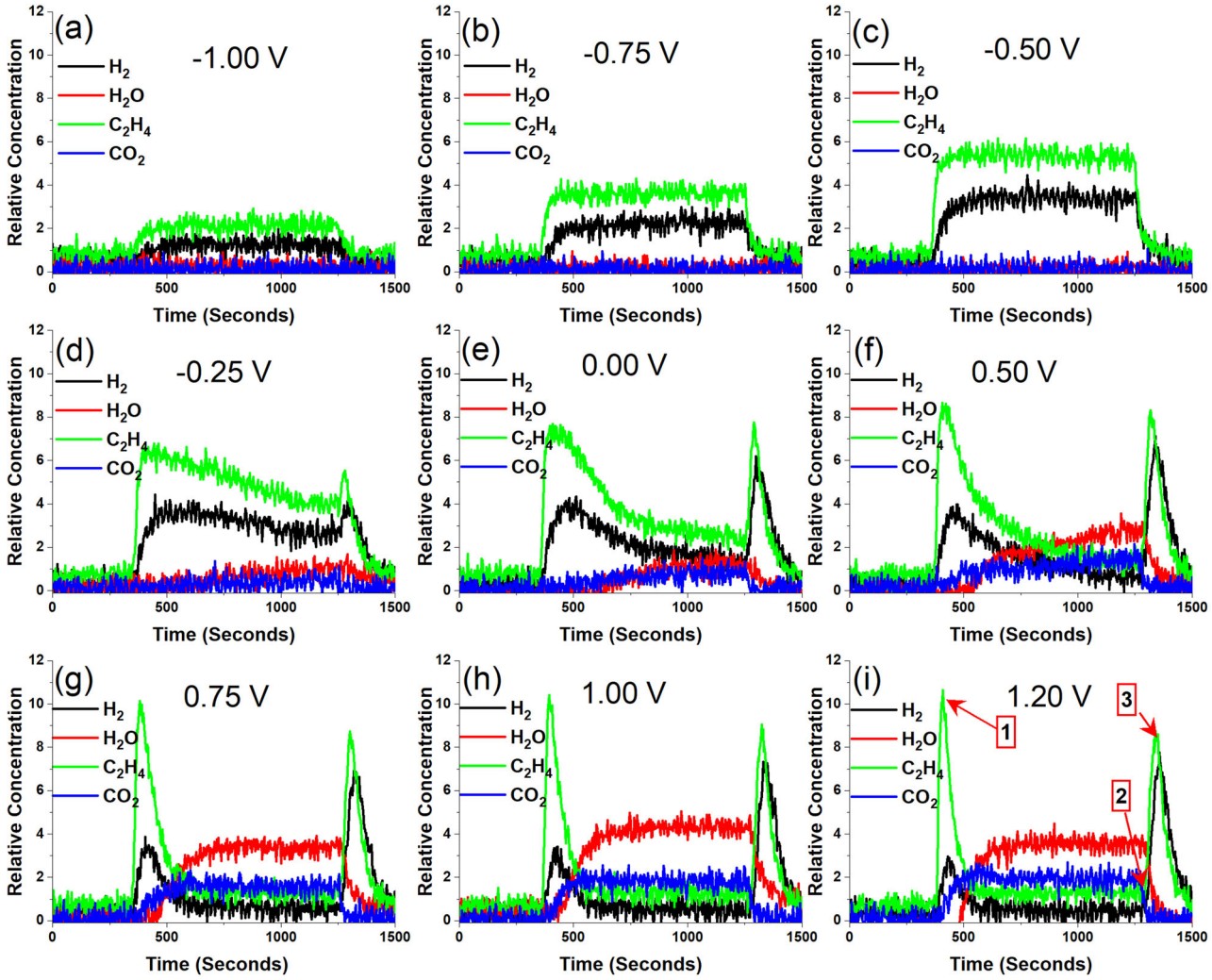

**Fig. 2 Mass spectroscopic analysis of outlet stream under chronoamperometric measurements.** Mass spectroscopic analysis of the outlet gas stream during chronoamperometric measurements observed under various bias potentials with respect to the reference electrode: **a** −1.00 V, **b** −0.75 V, **c** −0.50 V, **d** −0.25 V, **e** 0.00 V, **f** 0.50 V, **g** 0.75 V, **h** 1.00 V, and **i** 1.20 V. The black and red curves indicate $H_2$ and $H_2O$, while green and blue curves indicate $C_2H_4$ and $CO_2$, respectively.

overpotentials at the first point assuming a 2e process as per Supplementary Eq. (3) (a four-electron pathway provide a FE of 70%). This value is maintained throughout the duration of the measurement for low overpotentials. However, increasing the overpotential reduced the FE at both points with FE values reaching as low as 3% at applied potentials >0.5 V at point 2. This reiterates our observation on CV and CA measurements that low overpotentials help produce ethylene selectively while higher biases produce $CO_2$ and water. Since we observe both ethylene and hydrogen in the product stream with a little bit of CO, among the possible reactions involved in EC-OCM as shown in Supplementary Eqs. (3–8), Supplementary Eq. (3) involving two electrons seems to be the preferred process occurring under the EC-OCM conditions (for OCM reactions, see Supplementary Note 1). In addition to FE, we also calculated the carbon atom efficiency by measuring the consumed methane and produced ethylene and $CO_2$ that showed about 30–40% carbon efficiency. We further evaluated the impact of flow rates on the product distribution by carrying out CA measurements at two select potentials, −0.5 and 1.0 V, and at three different flow rates 50, 75, and 100 standard cubic centimeter per minute (SCCM). For the CA plots and corresponding mass spectroscopic results, see Supplementary Fig. 7. Lower flow rates help improve the

production of ethylene and hydrogen although a further inspection of the ratio between the peaks for ethylene and hydrogen indicates a higher production of hydrogen suggesting an increased contribution of CO along with ethylene as per Supplementary Eqs. (3) and (5) (for flow rate dependence results, see Supplementary Table 1 and Supplementary Fig. 7). The reduced flow rates and associated higher residence time of the reactants on the catalyst surface seem to facilitate the higher oxidation product, CO over ethylene. Nevertheless, the trends observed at 1 V in Fig. 2h are observed again with varying flow rates as the ability to produce higher oxidation products such as $H_2O$ and $CO_2$ at high overpotentials remained the same.

**Chemical stability of SFMO and formation of carbonates.** Nevertheless, the reason for loss of activity in the CV upon cycling is still unclear. Hence, to verify the chemical stability of these powders under methane conversion conditions, we exposed both powders (SFMO and SFMO-075Fe) to pure methane in a thermogravimetric analysis (TGA) set-up with a temperature window of 25 °C to 900 °C and a hold at 900 °C for 1 h. As depicted in Fig. 3a, SFMO and SFMO-075Fe both show substantial weight gain in methane environment (58 and 43%, respectively) at 900 °C. A possible explanation for this weight gain

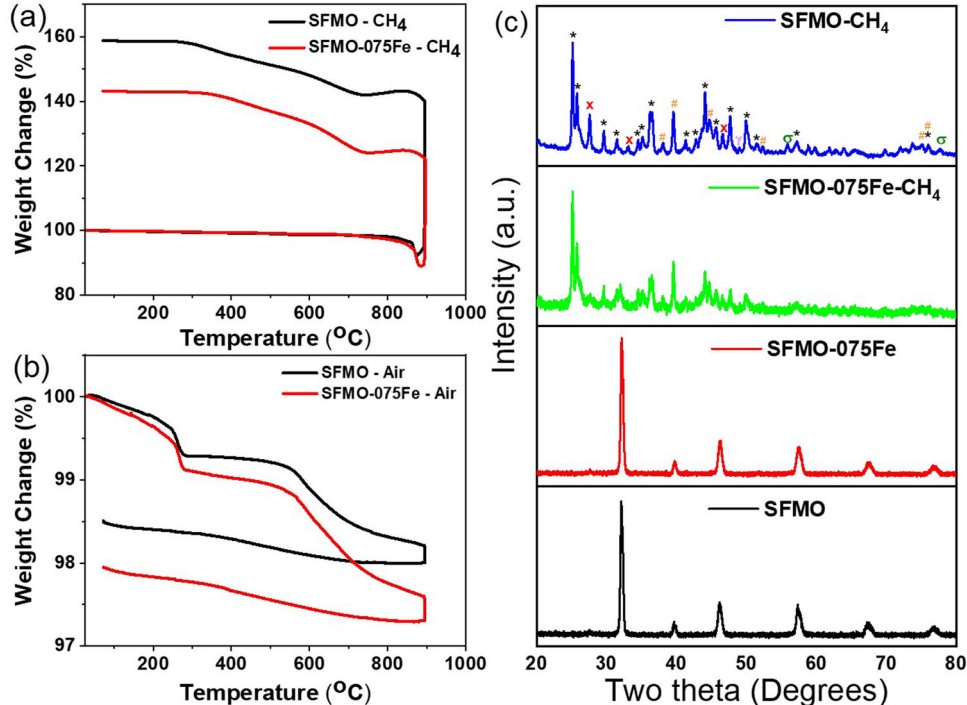

**Fig. 3 Chemical stability measurements. a** TGA plots obtained under pure methane atmosphere and **b** TGA plots obtained under air. The black and red curves indicate SFMO and SFMO-075Fe, respectively, in **a**, **b**. **c** PXRD of the as-prepared SFMO (black), SFMO-075Fe (red), and SFMO-075Fe and SFMO after TGA under methane (green and blue, respectively). The symbols indicate *—SrCO$_3$, X—SrMoO$_x$, #—MoC, γ—Fe, and σ—MoO$_x$.

could be the conversion of strontium in the SFMO into strontium carbonate. However, a complete conversion of all the Sr into SrCO$_3$ would only lead to about 21.83% weight gain even with an external oxygen supply (for SrO to SrCO$_3$ weight gain calculation, see Supplementary Note 2). This clearly indicates a possible carbide formation or coking during the CH$_4$ exposure at 900 °C. Nevertheless, no weight gain is observed in TGA measurements for both powders under air atmosphere and instead about ~2% weight loss is observed (Fig. 3b). Hence, such a huge weight gain in CH$_4$ environment could be explained by significant coking, carbide formation, carbonate formation, or by a combination of these three processes that would result in the disintegration of the crystal structure. Hence, we compared the powder X-ray diffraction (PXRD) patterns of both as-prepared and methane-exposed powders as presented in Fig. 3c to investigate the variation in crystal structure upon methane exposure. Both SFMO and SFMO-075Fe are prepared by a microwave combustion method as reported by Zhu et al. and the PXRD patterns matches well with the reported cubic perovskite structure with Fm3m space group[1,18,19]. However, PXRD of both powders after treated in methane shows the loss of all the characteristic peaks associated with the as-prepared SFMO. The peaks observed in the CH$_4$-treated samples could be associated with SrCO$_3$, SrMoO$_x$, MoC, MoO$_x$, and metallic Fe as identified in Fig. 3c. The peak corresponding to graphitic carbon could be hidden by the peaks associated with SrCO$_3$ in Fig. 3c (SFMO-CH$_4$). This clearly demonstrates the unsuitability of this material for application as a catalyst in EC-OCM. We also carried out temperature-programmed oxidation (TPO) measurements at three different gas combinations between methane and oxygen (100% CH$_4$:0% O$_2$, 95% CH$_4$:5% O$_2$, and 90% CH$_4$:10% O$_2$). In all three experiments, significant weight gain was observed on the SFMO powders as presented in Supplementary Table 3 (73, 101, and 16%, respectively) (for TPO weight gain details see Supplementary Table 3, PXRD results see Supplementary Fig. 9, and mass

spectra results see Supplementary Fig. 10a–c). Maximum weight gain was observed for 95% CH$_4$:5% O$_2$ mixture while the lowest weight gain of 16% was observed for 90% CH$_4$:10% O$_2$ gas mixture. PXRD patterns obtained for the three samples indicate loss of crystal structure in all three samples (Supplementary Fig. 9). Mass spectra obtained on the outlet stream for these measurements is given in Supplementary Fig. 10a–c. As shown in Supplementary Fig. 10a, under pure methane conditions, there is huge spike in H$_2$ production that indicates coke formation around 800 °C. In the case of 5% O$_2$ mix with CH$_4$, there is an initial surge in CO production around 400 °C followed by a big spike in hydrogen production similar to the pure methane scenario that indicates the onset of coking. Samples exposed to 100% CH$_4$ and 95% CH$_4$:5% O$_2$ recorded weight gains of 73.5 and 101%, which further confirms coking. In the case of 10% O$_2$ mixed with CH$_4$, such a sharp spike in H$_2$ production is not observed along with only a 16% weight gain indicating significantly reduced coking. These measurements prove the unsuitability of SFMO material for OCM or EC-OCM measurements and explains the reduction in peak size with time during CV measurements as shown in Fig. 1.

We next carried out X-ray photoelectron spectroscopic (XPS) measurements on the as-prepared and 100% CH$_4$-exposed SFMO powders to further understand the changes in SFMO material and the results are shown in Fig. 4. The survey scan in Fig. 4a clearly indicates an increase in the C 1s peak after CH$_4$ exposure in comparison to the as-prepared material. The percentage of carbon in the SFMO increased from few percentage points to 72% with CH$_4$ exposure, thus indicating significant carbon incorporation. Interestingly, the as-prepared SFMO shows a small amount of carbon as well that could be due to some residual SrCO$_3$ that did not burn away during the sintering process. This is further validated through the high-resolution Sr 3d$_{5/2}$ spectra. The Sr 3d$_{5/2}$ peak of as-prepared SFMO can be deconvoluted into two components as shown in Fig. 4c. Both SrO and SrCO$_3$ are present

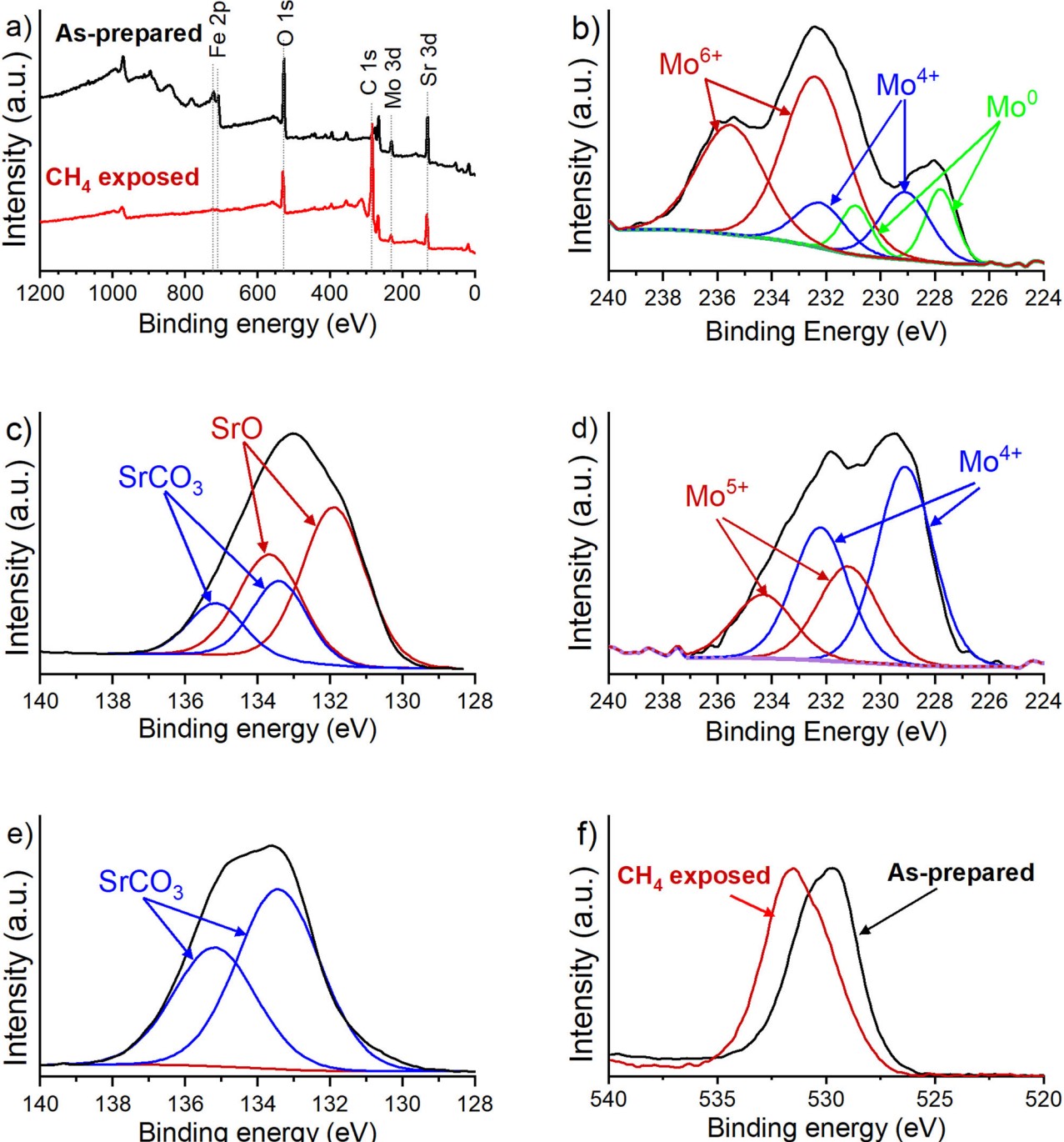

**Fig. 4 Analysis of carbonate formation. a** XPS survey scans for SFMO as-prepared and after TGA in pure methane, **b** Mo $3d_{5/2}$ peak obtained for as-prepared SFMO, **c** Sr $3d_{5/2}$ peak for as-prepared SFMO, **d** Mo $3d_{5/2}$ peak obtained for SFMO after TGA in $CH_4$, **e** Sr $3d_{5/2}$ peak for SFMO after TGA in $CH_4$, and **f** oxygen O 1s peak for SFMO before and after TGA in $CH_4$.

with the $3d_{5/2}$ bands located at 132.9 and 133.4 eV, respectively. The as-prepared sample shows the presence of $SrCO_3$ although the composition is dominated by SrO. However, after $CH_4$ exposure, the Sr-O dominant structure of SFMO is transferred to a Sr-$CO_3$ dominant structure further demonstrating the formation of carbonate during the $CH_4$ exposure process as shown in Fig. 4e. This is also demonstrated by the change in peak position for the O 1s. The peak shifted from a highly metal–oxygen bond orientation to carbon–oxygen bond orientation as shown in Fig. 4f. The carbonate formation is a common problem for alkaline earth-based high temperature electrode and electrolyte

materials since they form their respective metal carbonates that are thermodynamically the preferred product[12,13]. This is further confirmed by the HSC Chemistry calculations between SrO and $SrCO_3$ as shown in Supplementary Fig. 1 (for HSC data, see Supplementary Fig. 1). The high-resolution Mo $3d_{5/2}$ spectra is shown for the as-prepared and $CH_4$-exposed samples in Fig. 4b, d, respectively. The as-prepared sample consists of primarily $Mo^{6+}$, where the $3d_{5/2}$ energy band is located at 232.4 eV. After $CH_4$ exposure, the Mo $3d_{5/2}$ peak can be deconvoluted into two peaks that are consistent with a mixture of 5+ and 4+ oxidation states with $3d_{5/2}$ bands at 231.2 and 229.1 eV, respectively. The

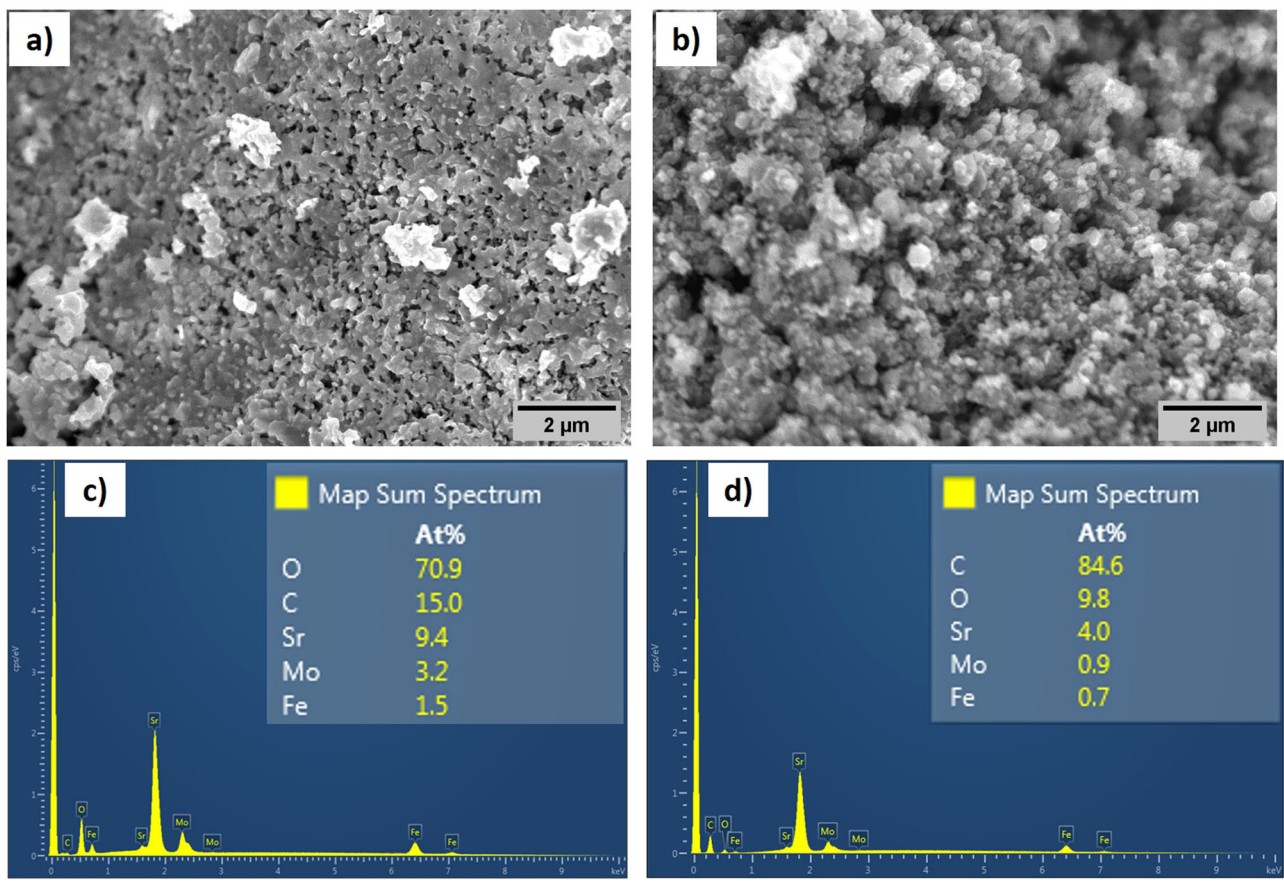

**Fig. 5 Microscopic analysis of the surface.** SEM images obtained for SFMO **a** before and **b** after $CH_4$ treatment indicating marked differences in particle agglomeration. **c, d** EDX elemental quantification for SFMO before and after $CH_4$ treatment, respectively, showing an increase in carbon content from 15 to 84 At%.

shift from a mostly 6+ oxidation state in the as-prepared SFMO to 4+ oxidation state in the $CH_4$-exposed sample along with significant contribution from 5+ oxidation state indicates the possible formation of $MoO_xC_y$ and $SrMoO_x$[20]. This is well in accordance with the PXRD results where peaks corresponding to $SrMoO_x$ and MoC were observed. HSC Chemistry calculations also indicate $SrMoO_4$ as thermodynamically favored product. Further, once formed, $SrMoO_4$ will not react with more carbon sources to produce $SrCO_3$ and $MoO_3$ as this reaction is unfavored at this temperature (Supplementary Fig. 1). As a result, both $SrCO_3$ and $SrMoO_4$ are the major decomposition products observed in PXRD (Fig. 3c). We also observed that in the XPS survey scans the peak intensities corresponding to the metal components in the $CH_4$-treated samples decrease in line with carbon deposition, especially in the case of Fe. Among the metal components of SFMO, iron is known to catalytically cleave the C–H bonds in methane under non-oxidative conditions. This property has been utilized to grow carbon nanotubes using iron nanoparticle-based catalysts with methane as a carbon source[21,22]. Thus, the observation of weight gain in TGA coupled with significant reduction in peak intensity for the Fe 2p peak in XPS could indicate the formation of carbon layers on top of the Fe. XPS quantification of the peak intensities presented in Supplementary Table 3 indicate a rise in C1s peak from 11.64 to 76.50% while contribution from all other components reduced significantly. This is further supported by the observation of carbon in XPS and PXRD measurements for the SFMO powders treated with $CH_4$ during the TGA measurements (Figs. 3a, c and 4a).

The scanning electron micrographs (SEM) obtained for as-prepared and $CH_4$-treated SFMO are presented in Fig. 5a, b along with their respective energy-dispersive X-ray (EDX) plots in Fig. 5c, d. As-prepared SFMO shows the presence of a granular structure with grains in the micron-size range. The EDX show the presence of Sr, Fe, and Mo along with carbon that could be due to unburnt carbonates. However, upon exposure to methane at 900 °C, the carbon contribution increased from 15 to 84 At% indicating a clear carbon deposition. This is in accordance with XPS quantification measurements where the carbon content increased from about 11% in as-prepared SFMO-075Fe to 76% in the $CH_4$-treated SFMO-075Fe. Further, the SEM image also show surface deposits after methane treatment. An elemental mapping of as-prepared and methane-treated SFMO-075Fe samples indicate a clear rise in carbon concentration on the surface specifically on these surface deposits (for elemental mapping images, see Supplementary Fig. 11 (as-prepared) and Fig. 12 (methane treated)). Our results shown here clearly demonstrate the disintegration of SFMO crystal structure along with significant carbon deposition. However, the activity centers in this catalyst responsible for methane activation are not clear at this point. The Fe nanoparticles exsolved in situ by reducing in 4% $H_2$ were claimed to be the active center with resistance toward coking by Zhu et al.[1]. However, a similarly exsolved $Sr_2FeMo_{0.65}$-$Ni_{0.35}O_{6-\delta}$ catalyst containing FeNi bimetal alloy nanoparticle was operated under humidified conditions to avoid coking in a solid oxide fuel cell anode where methane is completely oxidized to $CO_2$ and $H_2O$ under a higher flux of oxide ions[23]. This agrees with our observation of $CO_2$ and $H_2O$ at higher applied biases.

Moreover, Fe nanoparticle-based catalysts have been reported to suffer from coking and catalysts with single site Fe with no adjacent Fe sites are essential to achieve resistance toward coking[7,24]. In terms of durability, while an hour of exposure to $CH_4$ at 900 °C was sufficient to decompose the crystal structure, SFMO-based electrodes show sustained performance for multiple CV cycles at 1 mV s$^{-1}$ spanning about 10 h. However, as seen in Fig. 1, the catalytic activity was on a decline with increasing cycle numbers while background-level ethylene production was on the rise. $SrCO_3$ mixed with SrO is known for OCM catalysis where the ratio of SrO to $SrCO_3$ was varied. The Sr-O-rich catalyst was reported to provide higher $C_2$ yield in comparison to a $SrCO_3$-rich catalyst composition[25]. Similarly, MoC and molybdenum oxycarbides ($MoO_xC$) are known to catalyze the OCM[26]. Thus, the decrease in EC-OCM catalytic activity could be explained by the formation of $SrCO_3$, whereas the decomposition products such as $MoO_xC$ and MoC may increase the regular OCM reaction. Nevertheless, our results clearly demonstrate the advantage of EC-OCM in inducing product selectivity as a low overpotential window of −0.75 to −0.25 V in Figs. 1 and 2 was shown to produce ethylene preferentially. This observed potential window is in line with the reaction equilibrium cell potential window of −0.6 to −0.8 V for ethylene production calculated from HSC Chemistry calculations (Supplementary Fig. 6). The $CH_4$ to $C_2H_4$ transition is accepted to happen via methane adsorption on the catalyst surface as the first step followed by formation of $CH_3$ radicals that reacts with other $CH_3$ radicals in a gas phase coupling step to produce $C_2$ products[27]. However, with SFMO, $CH_4$ is shown to adsorb at the interface between Mo and Fe clusters. After the first C-H bond cleavage, $CH_3$ radicals remain adsorbed on the adjacent Fe clusters and continues to take part in C-H bond cleavages[1]. This type of adsorption explains the propensity to form coking. However, the charge transfer from adsorbed $CH_4$ molecule to catalyst surface is also essential for methane activation[2]. The $CH_4$ molecule adsorbed on SFMO can continuously donate electrons coupled with various active surface-bound oxygen species under applied electrochemical potentials where the applied biases determine the incoming oxygen flux and their oxidative power. Thus, as observed from Fig. 1, we believe at low applied potentials the methane activation is resulting in partial oxidation products such as ethylene while at higher biases the product stream is dominated by complete oxidation products, such as $CO_2$ and $H_2O$.

## Conclusion

We evaluated SFMO-075Fe for electrochemical activation of methane to produce ethylene after its ground-breaking performance as reported by Zhu et al.[1]. While Zhu et al. demonstrated high performance at applied potentials of 1.2–1.6 V, our results indicated significant $CO_2$ and $H_2O$ production under these conditions. Interestingly, Zhu et al. did not observe either of these two in their measurements. Further, CV measurements carried out in a wide operating window combined with mass spectra analysis demonstrate that methane could be activated at very low applied biases with respect to the cathode acting as reference electrode. Nevertheless, the chemical stability of SFMO-075Fe under methane activation conditions seem to be inadequate for meeting EC-OCM durability requirements. This is confirmed by the significant weight gain (>40%) in TGA analysis under methane environment at 900 °C coupled with XPS and XRD results that indicate the formation of $SrCO_3$ and carbon in the methane-exposed SFMOs. Further measurements at various $CH_4$ to $O_2$ ratios in TPO measurements revealed once again the tendency of this material to produce coke up to 10% $O_2$ in the reaction mixture stream. Hence, this material may not be suitable

for long-term methane conversion application. Further research is required to find chemically stable and durable catalysts. However, this study demonstrates the unique ability of EC-OCM to control the oxidation and as a result improve selectivity. Our study also provides a pathway to fine tune the operating conditions between partial oxidation and complete oxidation that would help mitigate coking problems in methane conversion experiments.

## Methods

**Materials**. Precursor materials, such as $Sr(NO_3)_2$, $Fe(NO_3)_3.9H_2O$, $(NH_4)Mo_7O_{24}.4H_2O$, poly(vinyl alcohol) (PVA), citric acid, $La_{0.9}Sr_{0.1}Ga_{0.8}Mg_{0.2}O_{3-\delta}$ (LSGM), $Ce_{0.8}Gd_{0.2}O_2$ (GDC), and silver wire, were all procured from Millipore Sigma®. Silver mesh current collector, high temperature sealing paste (CAP-552), thinner for high-temperature sealing paste (CAP-552-T), alumina slurry, and alumina felt seals were purchased from Fuel Cell Materials®. Ultra high purity (UHP) $CH_4$ and 3.9% $H_2$ balanced in $N_2$ were purchased from Airgas®.

**Preparation**. Fe-doped SFMO powders were prepared by microwave-assisted combustion method[1]. Appropriate amounts of (for 2 mM of SFMO) $Sr(NO_3)_2$, $Fe(NO_3)_3.9H_2O$, and $(NH_4)Mo_7O_{24}.4H_2O$ were mixed in hot deionized water. One gram of PVA was added with constant stirring until a dark red suspension was obtained. To this suspension, 1 g of citric acid was added gradually and the stirring was continued for another 15 min. The mixture was transferred to a bigger beaker (from 50 to 250 ml) and combusted in a microwave chamber. The obtained black and yellowish powders were grounded using a pestle and mortar and calcined at 1000 °C for 5 h at 3 °C min$^{-1}$ to obtain black-colored pure-phase SFMO powders. Commercially obtained LSGM powders were used for electrolyte membrane fabrication. In all, 1.6 g of LSGM powder was placed in a 25 mm pellet pressing die and uniaxially pressed at 1000 psi for 3 min. The obtained pellet was iso-statically pressed at 240 MPa for 3 min. The obtained pellets were sintered at 1175 °C for 12 h at 5 °C min$^{-1}$. The diameter and thickness of the sintered pellet was 20 and 1 mm, respectively. The shrinkage in diameter for the LSGM pellet was 20%.

**Electrochemical experiments**. For electrode preparation, SFMO and GDC powders were mixed with a 65:35 ratio and dispersed in α-terpineol with appropriate amount of cellulose added as pore former. GDC was included to enhance the triple-phase boundary in the electrode layer[28]. Previous reports on SFMO compatibility with a variety of materials indicated that SFMO forms a secondary layer only with YSZ[18]. The ratio of 65:35 was maintained to keep reproducibility with Zhu et al.[1]. The resultant mixture was ultrasonically mixed for an hour to obtain a slurry. The slurry was brush coated on the LSGM pellet with an active electrode area of 1 cm$^2$ and heat treated at 1100 °C for 3 h in air. SFMO was used as cathode and SFMO-075Fe was used as anode. Silver mesh current collectors were attached on the electrodes using silver paste and dried for a minimum of 20 min. Thus, prepared cell was placed on an alumina tube and sealed gas tight using alumina felt seal and high-temperature sealing paste. The cell set-up was dried in air at room temperature for 4 h followed by in situ heat treatments for 2 h each at 95 and 260 °C followed by 30 min sintering at 550 °C. Finally, the electrodes were reduced in 4% $H_2$ balanced in $N_2$ at 800 °C for an hour after which pure methane was supplied to the anode while air was supplied to the cathode. The flow rates were maintained at 100 SCCM. The outlet gas stream was continuously monitored by an MKS Cirrus 2 mass spectrometer and also periodically by an SRI 8610C gas chromatograph. Electrochemical experiments were carried out using a Gamry Reference 600 Potentiostat. TGA experiments were conducted using a TA Instruments SDT Q600. Measurements were taken in air or pure methane with a flow rate of 50 ml min$^{-1}$ using a heating rate of 5 °C min$^{-1}$ from 25 to 900 °C and held at 900 °C for 1 h. XPS measurements were performed on a Kratos Ultra DLD spectrometer using a monochromatic Al Kα source operating at 150 W (1486.6 eV). The operating pressure was $5 \times 10^{-9}$ Torr. Survey spectra were acquired at a pass energy of 160 eV and high-resolution spectra were acquired at a pass energy of 20 eV. XPS data were processed using the Casa XPS software. X-ray diffraction measurements were done in a PANalytical Xpert Pro instrument using Cu Kα radiation, in the range $2\theta = 20$–80°, and operating at 40 kV and 40 mA on a zero-background holder. SEM-EDX measurements were carried out on Hitachi S-5200.

**TPO measurements**. TPO measurements were carried out in a tubular furnace set-up. Approximately 200 mg of SFMO-075Fe was placed in an alumina boat and placed inside a quartz tube, which was set in the tubular furnace. The quartz tube was supplied with various mixtures of gases, 100% $CH_4$, 95% $CH_4$:5% $O_2$, and 90% $CH_4$:10% $O_2$ at a flow rate of 100 SCCM. The outlet of the quartz tube was connected to a mass spectrometer and analyzed. The samples were heated at 5 °C min$^{-1}$ to 900 °C and held for 1 h followed by cooling to room temperature at 5 °C min$^{-1}$.

## Data availability

The datasets generated and analyzed during the current study are available from the corresponding author on reasonable request.

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

## Acknowledgements

The authors thank the NSF – Center for Innovative and Strategic Transformation of Light Alkane Resources (CISTAR) (NSF – CISTAR award number EEC-1647722) for funding this work.

## Author contributions

K.P.R. and L.H.D. conducted the preparation of materials and electrochemical experiments. A.B. carried out the characterization experiments including the analysis of XPS. F.H.G. supervised this work. All authors were involved in the data analysis and discussion.

## Competing interests

The authors declare no competing interests.
