## [Peer Review File · Communications Chemistry]

Reviewers' comments:

Reviewer #1 (Remarks to the Author):

The work by Ramaiyan et al. is an interesting consideration of SFMO for EC-OCM. Motivated by a recent report (reference 1, Zhu et al. Nat Com 2019), the authors consider the performance of SFMO-075Fe in a solid oxide electrolyzer for EC-OCM, detecting products online by mass spec. Interestingly, the authors observe very different product distribution compared to Zhu et al. at the same potentials (vs. counter electrode): Ramaiyan observe C₂H₄ and H₂ primarily at negative biases, instead observing complete oxidation to CO₂ and H₂O at the 1.2+ volts (where Zhu et al. observed C₂H₄ + H₂). The authors have some nice detailed characterization of the material after thermal exposure to CH₄ atmosphere (albeit without applied potential and associated oxygen anion flux). However, while interesting in the differences between works, without additional insight into this process the present work reads as a more specialized contribution rather than an urgent communication across chemical sciences. Furthermore, the authors should consider further the following aspects of their work:

- Re: Figure S7 and the calculation of faradaic efficiency, no water is observed (product of equal stoichiometry to H₂ in eqn S3). How can this be explained?
- Re: HSC calculations in Fig S1 and S6, these are thermochemical equilibria. What about the relevant equilibrium to the solid oxide electrolyzer, where you additionally control the (electro)chemical potential of oxygen via an applied voltage?
- Re: formation of carbonates and coking, your thermal treatment in pure methane atmosphere in the TGA and subsequent characterization similarly only represents the thermochemical equilibria as in your HSC calculations, but does not reflect the oxidizing contribution from the flux of oxygen anions through the lattice driven by the applied voltage. Thus, it is unclear how relevant such conclusions are in SOEC performance.

Reviewer #2 (Remarks to the Author):

Conversion of methane into ethylene in a heterogeneous catalytic process is a very important topic; however, there are significant challenges and problems remained. The low methane conversion and ethylene selectivity are normally not high enough, and the performance degradation is always present due to coking and sintering of catalyst nanoparticles. In this manuscript, the authors report effective activation of methane to ethylene with SFMO-075Fe catalyst in a solid oxide electrolysis process. The manuscript is acceptable for publication after some minor revisions listed below:

1. The effect of flow rate on the production of ethylene, water and carbon dioxide should be given.
2. The EDX plots, as a qualitative analysis tool, cannot be used for quantitative analysis, so it cannot be concluded that the increase of carbon contribution leads to clear carbon deposition.
3. Will GDC have a significant impact on methane oxidation in this work?
4. Are the samples for TGA tests pretreated? Normally, the air atmosphere causes the oxidation of the samples, which increases the mass, however, the authors found just the opposite. I would

recommend a test of temperature programmed oxidation coupled with mass spectroscopy. It is a useful and effective method to determine the amount of coke.

5. The authors should calculate the carbon atom balance. Atomic efficiency is also a very important parameter to evaluate methane conversion. For the reported work in reference 1, there could be carbon loss in the test.

6. I would recommend the authors to discuss the impact of microstructure of electrode on the conversion of methane. And some discussion is also needed for the chemical stability of SFMO electrode. The formation of SrCO_3 is an adverse impact on the methane activation but the phase segregation is also related to the crystallinity.

7. I don't agree with the authors on the calculation of over oxidation of methane under different potentials. Actually, in anode the oxygen species in the form of active states are the key species to activate C-H bond in methane. I think the conversion of methane to ethylene is not an electrochemical process but a simultaneous process that involves activation CH_4 to CH_3/CH_2 and gaseous coupling of CH_3/CH_2 to ethylene.

8. I would recommend the authors to add some discussion on the active oxygen species that evolved from lattice under driving potentials. The external potentials could change the oxygen species toward methane activation.

Point by point response to reviewers' comments

The reviewer comments are given in red while our response to the reviewers' comments are given in blue.

Reviewer #1 (Remarks to the Author):

The work by Ramaiyan et al. is an interesting consideration of SFMO for EC-OCM. Motivated by a recent report (reference 1, Zhu et al. Nat Com 2019), the authors consider the performance of SFMO-075Fe in a solid oxide electrolyzer for EC-OCM, detecting products online by mass spec. Interestingly, the authors observe very different product distribution compared to Zhu et al. at the same potentials (vs. counter electrode): Ramaiyan observe C_2H_4 and H_2 primarily at negative biases, instead observing complete oxidation to CO_2 and H_2O at the 1.2+ volts (where Zhu et al. observed $C_2H_4 + H_2$). The authors have some nice detailed characterization of the material after thermal exposure to CH_4 atmosphere (albeit without applied potential and associated oxygen anion flux). However, while interesting in the differences between works, without additional insight into this process the present work reads as a more specialized contribution rather than an urgent communication across chemical sciences. Furthermore, the authors should consider further the following aspects of their work:

We thank the reviewer for the positive view on our manuscript. We have included additional details in the revised manuscript by means of temperature programmed oxidation measurements of SFMO-075Fe to mimic the EC-OCM environment. HSC calculations have been utilized to deduce the cell EMF values and is incorporated in the revised manuscript along with a discussion. Our detailed responses to the individual comments raised by the reviewer is given below. We hope the reviewer concerns are adequately addressed in the revised manuscript.

1. Re: Figure S7 and the calculation of faradaic efficiency, no water is observed (product of equal stoichiometry to H_2 in eqn S3). How can this be explained?

We thank the reviewer for pointing this out. Water was observed at high applied biases but not observed at low applied biases. This is possibly due to instrumentation challenges as the water condenses in the tubes between the tubular furnace and the mass spectrometer. Further, quantifying produced water is also hindered by this condensation. Hence, we attribute the excess current that is unaccounted by the gaseous products to water. A statement reflecting this has been included in the revised manuscript.

2. Re: HSC calculations in Fig S1 and S6, these are thermochemical equilibria. What about the relevant equilibrium to the solid oxide electrolyzer, where you additionally control the (electro)chemical potential of oxygen via an applied voltage?

HSC calculations involving oxygen molecule as oxygen source was done to show the thermochemical equilibria. As per the reviewer's suggestion, we calculated the cell EMF values using the Gibbs reaction free energies as a function of operating temperature and is now given in the supplementary information of the revised manuscript (Figure S6b). However, carrying out a detailed DFT studies on the role of applied electrochemical potential is beyond the scope of this manuscript. Nevertheless, our cyclic voltammetry results clearly demonstrate the role of applied voltage in controlling the electrochemical potential as observed from the specific windows for specific product selectivity.

3. Re: formation of carbonates and coking, your thermal treatment in pure methane atmosphere in the TGA and subsequent characterization similarly only represents the thermochemical equilibria as in your HSC calculations, but does not reflect the oxidizing contribution from the flux of oxygen anions through the lattice driven by the applied voltage. Thus, it is unclear how relevant such conclusions are in SOEC performance.

We agree with the reviewer on the oxidizing contribution from the oxygen flux. To address this, we carried out temperature programmed oxidation measurements where the SFMO-075Fe powders were exposed to varying methane to oxygen ratios (100:0, 95:5, and 90:10) in the temperature range of 25 - 900°C and monitored the outlet gas stream and also measured the weight changes. Our results show a clear disintegration of the crystal structure along with significant weight gain in all the three measurements. As with TGA measurements, we also observed significant coking in all three TPO measurements. The PXRD patterns and TPO graphs are now included in the supporting information of the revised manuscript (Figures S9 and S10).

Reviewer #2 (Remarks to the Author):

Conversion of methane into ethylene in a heterogeneous catalytic process is a very important topic; however, there are significant challenges and problems remained. The low methane conversion and ethylene selectivity are normally not high enough, and the performance degradation is always present due to coking and sintering of catalyst nanoparticles. In this manuscript, the authors report effective activation of methane to ethylene with SFMO-075Fe catalyst in a solid oxide electrolysis process. The manuscript is acceptable for publication after some minor revisions listed below:

We thank the reviewer for his positive comments. We have carried new experiments and modified the manuscript to include new information in order to address the reviewer comments. Our point by point response to the reviewer comment is given below.

1. The effect of flow rate on the production of ethylene, water and carbon dioxide should be given.

The effect of flow rates on the product distribution has been analyzed by chronoamperometric measurements at three different flow rates 50, 75 and 100 sccm at two select potentials representative of low and high applied biases viz., -0.5 V and 1.0 V. We observed that the lower flow rates help increase the ethylene production. However, possibly due to the higher residence time of the reactants, CO is also observed increasingly at lower flow rates. The results are now incorporated in the supporting information of the revised manuscript along with a discussion in the main manuscript.

2. The EDX plots, as a qualitative analysis tool, cannot be used for quantitative analysis, so it cannot be concluded that the increase of carbon contribution leads to clear carbon deposition.

We thank the reviewer for pointing this out. We carried out XPS measurements on the samples before and after exposure to methane to quantify the carbon deposition along with EDX measurements. The quantification details from the XPS measurements which reveal similar increments in carbon content are now included in the revised manuscript along with the EDX measurements to support our claims. Both XPS and EDX measurements reflect the same trend where the carbon content increases after exposure to methane at 900 °C.

3. Will GDC have a significant impact on methane oxidation in this work?

GDC is incorporated to improve the triple phase boundary. GDC has been utilized to enhance the TPB of SFMO based electrodes (Das et al., *Electrochimica Acta* 354 (2020) 136759). Similarly, the chemical compatibility of SFMO with LSGM and SDC has been verified in the literature earlier (Liu et al., *Adv. Mater.* 2010, 22, 5478–5482). We have used a similar composition used by Zhu et al where higher currents have been achieved and hence we did not modify this process (Zhu et al., *Nature Communications* 2019, 10, 1173). This information is now included in the revised manuscript.

4. Are the samples for TGA tests pretreated? Normally, the air atmosphere causes the oxidation of the samples, which increases the mass, however, the authors found just the opposite. I would recommend a test of temperature programmed oxidation coupled with mass spectroscopy. It is a useful and effective method to determine the amount of coke.

We thank the reviewer for this insightful comment. The samples used in TGA are used as prepared. The preparation process involves sintering the sample at 1000°C. Hence, we did not carry out any further pretreatment before TGA measurements. We carried out TPO measurements as suggested by the reviewer. TPO measurements on the SFMO-075Fe samples have been carried out with varying methane to oxygen ratio (100:0, 95:5, and 90:10). The results clearly demonstrate the collapse of crystal structure along with significant weight gain as shown in Figure S9. Further, all the three measurements resulted in significant amount of coking as shown in Table S2 and Figure S10 in the revised supporting information. New discussion has been included in the revised manuscript to reflect these new results.

5. The authors should calculate the carbon atom balance. Atomic efficiency is also a very important parameter to evaluate methane conversion. For the reported work in reference 1, there could be carbon loss in the test.

We thank the reviewer for suggesting this calculation. We carried out carbon atom balance calculations that showed carbon efficiency in the range of 30 – 40%. However, methane consumption was less than five percent making accurate calculations a challenge. Nevertheless, the low carbon atom efficiency is in line with our results from TGA and TPO measurements where coking is observed under pure methane and methane with 10% oxygen conditions.

6. I would recommend the authors to discuss the impact of microstructure of electrode on the conversion of methane. And some discussion is also needed for the chemical stability of SFMO electrode. The formation of SrCO₃ is an adverse impact on the methane activation but the phase segregation is also related to the crystallinity.

We thank the reviewer for this suggestion. We have included a discussion on the CH₄ adsorption on the SFMO structure and its expected activation mechanism. We have included new results from temperature programmed oxidation under various methane to oxygen ratio that again reflect the poor chemical stability of SFMO-075Fe under the operating conditions of EC-OCM. PXRD results demonstrate the loss of crystallinity after exposure to methane. Our results indicate that this material is chemically not stable under EC-OCM conditions.

7. I don't agree with the authors on the calculation of over oxidation of methane under different potentials. Actually, in anode the oxygen species in the form of active states are the key species to activate C-H bond in methane. I think the conversion of methane to ethylene is not an electrochemical

process but a simultaneous process that involves activation CH_4 to CH_3/CH_2 and gaseous coupling of CH_3/CH_2 to ethylene.

The cyclic voltammetry results presented in Figure 1 indicate a specific electrochemical potential window for the activation of methane towards ethylene. Theoretical analysis by Zhu et al (Zhu et al., *Nature Communications* 2019, 10, 1173) indicate that CH_4 is adsorbed on the interface between Fe clusters in the SFMO system that is key for methane activation. We believe that this surface bound methane interaction with incoming oxide ions results in various oxidation products. While the low overpotential lead to a partial oxidation product, ethylene, at higher positive biases, over oxidation products such as CO_2 and H_2O is formed due to higher oxidation power and a higher flux of oxide ions. This discussion is now included in the revised manuscript.

8. I would recommend the authors to add some discussion on the active oxygen species that evolved from lattice under driving potentials. The external potentials could change the oxygen species toward methane activation.

We thank the reviewer for this suggestion. The electrochemical pumping of O^{2-} ions from the cathode to anode introduces various active oxygen species at the anode such as O^{2-} , O^- , O atom, and O_2 molecule as a charge rebalancing process happen at the anode. Previous reports indicate that the most active surface site to activate methane is O^- species (Palmer et al., “Periodic Density Functional Theory Study of Methane Activation over La_2O_3 : Activity of O^{2-} , O^- , O_2^{2-} , Oxygen Point Defect, and Sr^{2+} -Doped Surface Sites,” *J. Am. Chem. Soc.*, 124, 8452–8461, 2002 and Voskresenskaya et al., “Oxidant Activation Over Structural Defects of Oxide Catalysts in Oxidative Methane Coupling,” *Catal. Rev.*, 37, 101–143, 1995). A detailed discussion based on these earlier reports involving different oxygen species is now included in the revised manuscript.

REVIEWERS' COMMENTS:

Reviewer #1 (Remarks to the Author):

The authors have included notable revisions to the manuscript in light of the critique from both reviewers, and it is greatly improved by these changes. Their concerns were adequately addressed.

Reviewer #2 (Remarks to the Author):

I think the manuscript is acceptable for publication

Response to Reviewer comments

Reviewer #1 (Remarks to the Author):

The authors have included notable revisions to the manuscript in light of the critique from both reviewers, and it is greatly improved by these changes. Their concerns were adequately addressed.

We thank the reviewer for accepting our response and we are happy that the reviewer's concerns are adequately addressed.

Reviewer #2 (Remarks to the Author):

I think the manuscript is acceptable for publication.

We thank the reviewer for his positive opinion on our revised manuscript. Both the reviewer's comments has helped us improve our manuscript.